# Beautifying diffusion models: Learning context-aware filters for robust dense prediction on test-time corrupted images

## Abstract

Diffusion models have enabled input-based domain adaptation to unseen test-time corruption for the classification problem. Nevertheless, while dense prediction tasks share similar robustness issues with image-level classification, previous input adaptation work may fail to preserve the semantic information necessary for robust pixel-level prediction. To address the issue, we propose a novel diffusion-driven strategy that translates the corrupted inputs to the source domain (*i.e.*, the training data domain), while also preserving the semantic information (*i.e.* high-frequency shape information and low-frequency color information). We first studied how to leverage frequency filtering to guide the diffusion generation process and analyze the influence of different filters. From our experiments, we observed that utilizing both high and low spatial frequency information during diffusion driven denoising can substantially improve the adaptation performance of dense classification. This observation motivates us to develop a novel framework, *i.e.* a predictive frequency filtering-driven diffusion (FDD) adaptation, where we predict the filters from the corrupted test-time inputs and use them to condition the diffusion denoising process. We design a Y-like frequency prediction network to predict context-aware low-pass and high-pass filters. To train this network, we propose a novel data augmentation method, FrequencyMix, to generate pairs of clean and corrupted images. We validate our method via extensive experiments on two semantic segmentation datasets and two depth estimation datasets. Against a broad range of common corruptions, we demonstrate that our method is competitive with state of the art work.

## 1 Introduction

Distribution shift between the testing (target) and training (source) distributions poses a considerable challenge towards the scalable implementation of deep learning models (Filos et al. (2020),Rosche-witz et al. (2023)). Variations in illumination levels, weather conditions, and image quality Liang et al. (2023) cause distribution shifts between the training and test data. Existing work enhance the robustness of deep learning models to distribution shift either via model adaptation methods: by fine tuning the task models with the unlabeled target domain data (Wang et al. (2021); Prabhudesai et al. (2023)) or via input adaptation methods: by transforming the input target domain data to match the training data (Gao et al. (2023); Song & Lai (2023); Huang et al. (2023)).

The effectiveness of conditioning diffusion models (Zhang et al. (2023); Choi et al. (2021); Liu et al. (2023); Nichol et al. (2022); Dhariwal & Nichol (2021); Ho & Salimans (2022)) to generate user-defined outputs have motivated their use in denoising corrupted images (Gao et al. (2023); Choi et al. (2021); Wang et al. (2023b)). However, these methods apply predefined, image-level filters to condition the diffusion model. In this work, we aim to study the use of context-aware kernels to enhance the effectiveness of diffusion-driven denoising for dense classification tasks.

However, we observe that low-frequency information is insufficient for dense classification because crucial fine-grained edge information could be lost during denoising. Hence, diffusion-based TTA in dense prediction needs to preserve both the high frequency (edge information) and the low-frequency (*e.g.* color) information.

This observation drives our design to predict the **spatially adaptive pixel-wise kernels** from corrupted images for conditioning of the diffusion model. To address this problem, we propose a simple yet effective context-aware kernel prediction network to preserve semantic information during input adaptation. Our key insight is that the semantic information necessary for good performance in dense classification tasks comprises **high spatial frequency information** (*e.g.* edge information), and **low spatial frequency information** (*e.g.* texture information). Furthermore, since it is non-trivial to manually define robust frequency filters that effectively extract semantic information under varying noise conditions, we propose to train a Y-like Frequency Prediction Network(Y-FPN) to condition the diffusion model for input adaptation of dense classification tasks (semantic segmentation and depth estimation). To our best knowledge, this is the first work that incorporates learned spatially adaptive kernels to condition a denoising diffusion model for robust dense prediction.

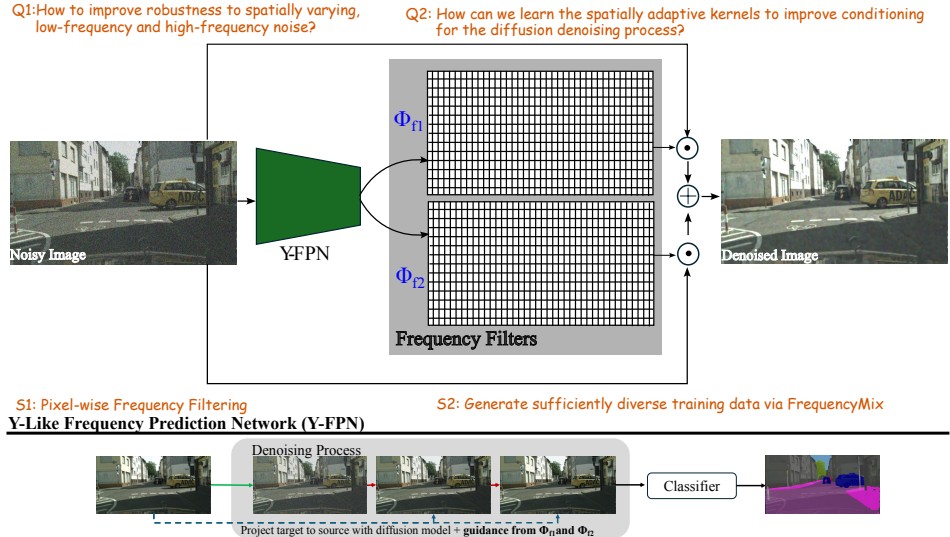

**Figure 1:** We show the workflow of our method, which was inspired by the questions, Q1 and Q2. In our approach, we introduce 2 solutions (S1 and S2) for diffusion-driven test-time input adaptation. $\Phi_{f1}$ and $\Phi_{f2}$ denote the spatially adaptive high-frequency and low-frequency filters. We apply the learned filters to condition the denoising process.

## 2 RELATED WORK

**Test-time domain adaptation.** Test Time Adaptation includes *model adaptation*, which involves finetuning the model during inference and *input adaptation*, which transforms the target domain data to resemble the source domain data. Kim & Byun (2020) observed that arbitrary style transfer-based approaches introduce image artifacts in the generated images which can hinder adaptation performance. Unlike style transfer methods and other GAN-based approaches (Hoffman et al. (2018)), diffusion models offer a greater degree of fine-grained control over the generated outputs (Nie et al. (2022); Gao et al. (2023); Song & Lai (2023); Huang et al. (2023)) to purify them.

Although numerous prior studies (Hu et al. (2021); Wang et al. (2021); Sivaprasad & Fleuret (2021); Wang et al. (2022); Prabhudesai et al. (2023); Shin et al. (2022)) have explored the integration of TTA into dense prediction tasks, existing work focus on fine tuning the task model with the target domain data. Recently, Prabhudesai et al. (2023) apply the diffusion modeling process for model adaptation. However, their method involves fine tuning both the diffusion model and the task model, which requires considerable computational resources. In contrast, we neither update the diffusion model nor the task model and our approach requires less computation during training. Instead of applying a predefined image-level filter for conditioning, we propose a Y-like Frequency Prediction Network (Y-FPN) that learns the high-frequency and low-frequency kernels for each pixel. These kernels can extract semantic information while being simultaneously robust to noise.

**Diffusion models.** Diffusion modeling has recently gained widespread prominence among deep generative models and demonstrates high generative capacity by iteratively refining inputs (Ho et al. (2020); Song et al. (2021); Song & Ermon (2019; 2020); Sohl-Dickstein et al. (2015); Blattmann et al. (2023); Nichol & Dhariwal (2021)). In essence, the idea behind diffusion, such as the denoising diffusion probabilistic model (DDPM) (Ho et al. (2020)), is to iteratively add noise to the data in the "forward process", followed by training a network to recover the original data in a "reverse process". Diffusion models also display a high degree of user controllability, allowing for conditioning on visual cues (Choi et al. (2021)), natural language (Liu et al. (2023); Nichol et al. (2022)), and even on class labels (Dhariwal & Nichol (2021); Ho & Salimans (2022)). Recent work (Gao et al. (2023); Oh et al. (2024)) have demonstrated the effectiveness of diffusion models in projecting non-Gaussian corrupted data towards the source distribution. In particular, Gao et al. (2023) show the effectiveness of a predefined low pass filter for robust test-time input adaptation of image classification tasks. Inspired by their work, we design our learning framework to selectively capture both high-frequency and low-frequency semantic information for test-time input adaptation involving dense classification tasks.

**Frequency-aware processing.** There is substantial ongoing interest in understanding deep learning from a frequency-modelling perspective and leveraging this understanding to achieve robust and generalizable vision systems (Chattopadhyay et al. (2023); Choi et al. (2021); Gao et al. (2023); Xie et al. (2023); Xu et al. (2023)). While it was earlier observed that classification models are biased towards colour and texture information, it was also observed that vision models make predictions based on a combination of low and high-frequency information (Yin et al. (2019); Wang et al. (2023a)). Notably, Choi et al. (2021) demonstrated the effectiveness of conditioning the diffusion model with predefined image-level frequency filters to enable user-defined image generation. Their work motivated diffusion-based test-time input adaptation for robust classification. Gao et al. (2023) employed predefined image-level filters to condition the pretrained diffusion model and preserve the image level information during denoising of the corrupted images. Different from existing work, we propose to learn the low-pass and high-pass kernels for conditioning diffusion-driven denoising as a defence against test-time corruptions.

## 3 METHODS

### 3.1 PRELIMINARY: LOW-PASS-FILTER-DRIVEN DIFFUSION ADAPTATION

We have a diffusion model pre-trained on a source dataset (*e.g.*, clean images). Given an input image $\boldsymbol{x}_0$ randomly sampled from a target distribution $q(\boldsymbol{x}_0)$, we aim to leverage the diffusion model to map $\boldsymbol{x}_0$ to the source domain without training. Specifically, an unconditional diffusion model comprises a forward process and a reverse process under a Markov chain. The forward process iteratively adds Gaussian noise to $\boldsymbol{x}_0$ over $T$ steps. Then, we obtain the distribution $q(\boldsymbol{x}_N|\boldsymbol{x}_0)$ with $N \in [1, \ldots, T]$. We sample from the distribution via $\boldsymbol{x}_N \sim q(\boldsymbol{x}_N|\boldsymbol{x}_0)$ and conduct reverse denoising to map $\boldsymbol{x}_N$ to the source domain (*e.g.*, training domain). Specifically, given the noisy image $\boldsymbol{x}_N^g = \boldsymbol{x}_N$, we aim to denoise it and get $[\boldsymbol{x}_{N-1}^g, \boldsymbol{x}_{N-2}^g, \ldots, \boldsymbol{x}_0^g]$ and $\boldsymbol{x}_0^g$ is desired to be within the source domain. The whole process estimates the joint distribution $p_\theta(\boldsymbol{x}_{0:N}^g) = p(\boldsymbol{x}_N^g) \prod_{t=1}^N p_\theta(\boldsymbol{x}_{t-1}^g|\boldsymbol{x}_t^g)$. with the conditional distribution as

$$p_\theta(\boldsymbol{x}_{t-1}^g|\boldsymbol{x}_t^g) = \mathcal{N}\left(\boldsymbol{x}_{t-1}^g; \boldsymbol{\mu}_\theta\left(\boldsymbol{x}_t^g, t\right), \boldsymbol{\Sigma}_\theta\left(\boldsymbol{x}_t^g, t\right)\right). \tag{1}$$

where $p(\boldsymbol{x}_N^g) := \mathcal{N}\left(\boldsymbol{x}_N^g; 0, \mathbf{I}\right), \boldsymbol{\Sigma}_\theta\left(\boldsymbol{x}_t^g, t\right) = \sigma_t \mathbf{I}$ are time-dependent constants, $\boldsymbol{\mu}_\theta$ is parameterized by a linear combination of $\boldsymbol{x}_t^g$ and $\boldsymbol{\epsilon}_\theta(\boldsymbol{x}_t^g, t)$ that are estimated from the network parameterized with $\theta$.

Gao et al. (2023) introduce a low-frequency preserving constraint to the reverse process. For the time step $t - 1$, we can summarize the steps of Gao et al. (2023) as: ❶ Sampling an example from the conditional distribution, *i.e.*, $\hat{\boldsymbol{x}}_{t-1}^g \sim p_\theta(\boldsymbol{x}_{t-1}^g|\boldsymbol{x}_t^g)$. ❷ Estimating the input image based on $\hat{\boldsymbol{x}}_{t-1}^g$ by

$$\hat{\boldsymbol{x}}_0^g = \sqrt{\frac{1}{\alpha_t}}\boldsymbol{x}_t^g - \boldsymbol{\epsilon}_\theta(\boldsymbol{x}_t^g, t)\sqrt{\frac{1}{\alpha_t} - 1}. \tag{2}$$

❸ Deriving the denoised image at $t - 1$ from the unconditional proposal $\hat{\boldsymbol{x}}_{t-1}^g$) and low-pass constraints,

$$\boldsymbol{x}_{t-1}^g = \hat{\boldsymbol{x}}_{t-1}^g - \boldsymbol{w}\nabla_{\boldsymbol{x}_t}\left\|\phi_{\mathsf{lp}}\left(\boldsymbol{x}_0\right) - \phi_{\mathsf{lp}}\left(\hat{\boldsymbol{x}}_0^g\right)\right\|_2, \tag{3}$$

where $\phi_{\text{lp}}(\cdot)$ is a spatially invariant, predefined low-pass filter, and $w$ controls the step size of the guidance. Then, we can iteratively update the estimate of the denoised image $x_{t-1}^g$ untill $t = 0$.

Intuitively, with Eq. (3), we try to balance the sample drawn using the pretrained model, *i.e.*, $\hat{x}_{t-1}^g \sim p_\theta(x_{t-1}^g | x_t^g)$ and the guidance (specified by the second term) that preserves the low-frequency semantic information. Gao et al. (2023) leverages this constraint to preserve the semantic information in the corrupted input image. *However*, a predefined image-level constraint that is effective for image level classification tasks cannot be directly used for dense prediction tasks due to the inherent difference in task complexity. The objective for image classification is to predict the category of the input image and is not as heavily reliant as dense prediction on pixel-level details for high accuracy predictions.

The recently demonstrated effectiveness of conditional diffusion models with predefined image-level filters (Gao et al. (2023); Choi et al. (2021); Wang et al. (2023b)) inspired us to study the effectiveness of frequency filters for conditioning diffusion models. We first hypothesize that low-pass filters are insufficient for dense classification tasks and include high-pass filters in our approach. We modified Equation (3),

$$x_{t-1}^g = \hat{x}_{t-1}^g - w\nabla_{x_t} \left[ \|\phi_{\text{lp}}(x_0) - \phi_{\text{lp}}(\hat{x}_0^g)\|_2 + \|\phi_{\text{hp}}(x_0) - \phi_{\text{hp}}(\hat{x}_0^g)\|_2 \right], \tag{4}$$

where $\phi_{\text{hp}}^c$ denotes a high-pass filter.

We then take the Cityscapes-C dataset and evaluate the effect of conditioning the diffusion model with predefined image-level frequency filters. To avoid introducing artifacts during frequency-based filtering, we apply a Hann window to reduce the magnitude of the transitions at the cutoff frequencies.

**Table 1:** Comparison of segmentation performance (mIoU) on Cityscapes-C via image-level frequency-driven conditioning of a pretrained diffusion model for image denoising. We apply predefined frequency cutoffs for both low-pass and high-pass filtering.

|  | Gauss. | Shot | Defocus | Glass | Zoom | Fog | Bright. | Elastic | JPEG. | Pixel |
|---|---|---|---|---|---|---|---|---|---|---|
| High Pass | 35.23 | 38.32 | 50.37 | 40.16 | 11.88 | 44.96 | 35.16 | 41.11 | 35.81 | 39.60 |
| Low Pass | 51.94 | 48.92 | 57.31 | 53.37 | 21.20 | 55.87 | 52.22 | 54.54 | 44.23 | 55.51 |
| Low + High Pass | 54.13 | 54.21 | 57.32 | 53.49 | 19.35 | 56.04 | 56.61 | 58.62 | 42.37 | 58.40 |

## 3.2 DISCUSSION AND MOTIVATION

Based on the findings in Table 1, we observe the following: ❶ combining a high pass filter and a low pass filter generally outperforms either filter during conditioning of diffusion driven denoising. As expected, combining a high-pass filter with a low-pass filter improved robustness to low-frequency corruptions (*e.g.* brightness) compared to a single low-pass filter. Additionally, we also observed improved robustness to high frequency noise (*e.g.* Gaussian and Shot Noise). ❷ Conditioning the diffusion model with a high pass filter yields poorer performance compared to a low pass filter. While combining both filters improves test time adaptation, we observe that the high-pass filter may reduce denoising effectiveness (*e.g.* Zoom Blur, JPEG compression). This is because the high pass filter failed to remove the noise, leading to suboptimal conditioning of the diffusion model. This indicates that much of the critical semantic information required for robust dense classification has low spatial frequencies. Clearly, a predefined, image-level high pass filter is inadequate for test time adaptation of dense classification tasks. To address this issue, we propose a novel context-aware frequency filtering method in Section 4.

## 4 PREDICTIVE FREQUENCY FILTERING-DRIVEN DIFFUSION

In this section, we discuss our technical contributions for advancing TTA on dense prediction. Firstly, we discuss ❶ Y-like Frequency Prediction Network, which assists the input translation process of the diffusion model in terms of feature constraints; and ❷ FrequencyMix Training to help the network learn the suitable high-pass and low-pass filters.

## 4.1 Y-LIKE FREQUENCY PREDICTION NETWORK FOR VISUAL FILTERING

In this section, we present our frequency-based filtering method for robust test-time input adaptation. Applying a predefined, image level filter can degrade performance, especially if the filter removes

semantic information or introduces noise. To achieve overall robustness to a wide variety of corruptions, we first consider the following challenge: *how to improve robustness to spatially varying, low-frequency and high-frequency noise?*

For example, corruptions such as "motion blur" vary spatially and disproportionately affect different parts of the image. While a straightforward solution would be to apply a uniform image-level low-pass filter to preserve the texture information, this may inadvertently remove the edge information required for dense classification. Furthermore, many real-world image corruptions are complex and comprise several different effects (blurring, color shifts and occlusion). For example, corruptions such as "glass blur" degrade the high-frequency information present in the image while simultaneously introducing low-frequency noise. Therefore, an ideal filter must be **spatially adaptive** and robust to noise across a broad frequency range. To address this challenge, we adopt a pixel-wise filtering approach, which is computationally efficient and effective in dealing with spatially varying noise, for test-time adaptation of dense classification tasks. Specifically, we process a corrupted image $\mathbf{I}^c \in \mathbb{R}^{H \times W}$ for pixel-wise filtering,

$$\mathbf{I}^f = \sum_{i=1}^{l=2} \mathbf{K}_i \odot \mathbf{I}^c, \tag{5}$$

where $\mathbf{I}^f \in \mathbb{R}^{H \times W}$ is the filtered image and $\odot$ denotes the pixel-wise filtering operation. $\mathbf{K}_i \in \mathbb{R}^{H \times W \times K^2}$ denotes the kernels of size $K \times K$ ($K = 3$) for the entire image. Based on our earlier findings demonstrating the effectiveness of combining a low-pass filter and a high-pass filter, we jointly apply $\mathbf{K}_{i=1,2}$ to the corrupted image $\mathbf{I}^c$ to generate $\mathbf{I}^f$.

Here, we are faced with the next challenge: *How can we learn the spatially adaptive kernels to improve conditioning for the diffusion denoising process?* Selecting an optimal frequency cutoff for the low-pass and high-pass filters is not trivial. The reason is that a single frequency cutoff is inflexible and does not guarantee robustness across the sheer diversity of corruptions. Manually selecting a frequency cutoff often improves performance for a limited set of corruptions while degrading performance for other corruptions. Furthermore, for dense classification tasks, manually choosing a frequency cutoff for each pixel is not tractable given the sheer number of pixels.

The successful application of Kernel Prediction Networks (KPN) across a variety of tasks, such as denoising (Mildenhall et al. (2018)), image inpainting (Li et al. (2022)) and deraining (Guo et al. (2021)), has encouraged us to leverage this versatile approach to learn the spatially adaptive kernels for conditioning diffusion models for dense classification tasks.

We propose to estimate the kernels $\mathbf{K}_{i=1,2}$ for conditioning the diffusion model from an arbitrary noisy image,

$$\mathbf{K}_{i=1,2} = \text{Y-FPN}(\mathbf{I}^c), \tag{6}$$

where Y-FPN denotes the Y-like frequency prediction network and shares a similar architecture with the UNet network (Ronneberger et al. (2015)). Unlike previous works (Mildenhall et al. (2018); Li et al. (2022); Guo et al. (2021)), which learn a single kernel for each task, Y-FPN predicts 2 separate filters per pixel for a given image. We derive the set of low-pass pixel-wise filters $\Phi_{lp}$ from $\mathbf{K}_{i=1}$ and apply the method by (Zou et al. (2023)) to constrain the weights of the low-pass filters. We obtain the set of high pass filters $\Phi_{hp}$ from $\mathbf{K}_{i=2}$ by similarly constraining the weights of the $\mathbf{K}_{i=2}$ and subsequently obtaining the difference between the filtered output and the input image.

Significantly, while we train the frequency prediction network via a denoising framework, we leverage the learned spatially adaptive filters to condition the pretrained diffusion model instead of directly applying the frequency prediction network to denoise the images. This allows us to leverage the existing generative capacity of the pretrained diffusion models to denoise a diverse range of image corruptions.

Since our test-time adaptation settings assume that no target domain data are available for training, we are faced with the challenge of acquiring suitable data to adequately and effectively train the frequency prediction network.

We apply the commonly used loss functions for image restoration *i.e.* $L_1$, SSIM (Structural Similarity) to train our network. We also include a Frequency Reconstruction Loss (Kim et al. (2021)), which is computed from the difference of both the denoised image $\mathbf{I}^f$ and the original clean image $\mathbf{I}$ in frequency space. We first apply the Fourier Transform to the denoised image $\mathbf{I}^f$ and the original clean image $\mathbf{I}$. We then compute the difference between Fourier transformed images and normalize

the result with a logarithmic function.

$$\mathcal{L}_{Freq}(\mathbf{I}^f, \mathbf{I}) = \log(1 + \frac{1}{HW}\|\mathcal{F}(\mathbf{I}) - \mathcal{F}(\mathbf{I}^f)\|), \tag{7}$$

where $H, W$ are the spatial dimensions of the image in frequency space. Bringing it all together, we have the following overall loss,

$$\mathcal{L}(\mathbf{I}^f, \mathbf{I}) = \|\mathbf{I}^f - \mathbf{I}\|_1 - \lambda_1 \text{SSIM}(\mathbf{I}^f, \mathbf{I}) + \lambda_2 \mathcal{L}_{Freq}(\mathbf{I}^f, \mathbf{I}), \tag{8}$$

where we set $\lambda_1 = 0.2, \lambda_2 = 0.1$.

## 4.2 FREQUENCYMIX TRAINING

In this part, we introduce our approach to generate sufficiently challenging data for training the Y-like Frequency Prediction Network (Y-FPN). Since Y-FPN must be robust to noise across all frequencies, we ❶ apply frequency-dependent perturbations and ❷ combine these frequency-dependent perturbations via **FrequencyMix** to increase the training data diversity.

**Frequency-Dependent Perturbations** Inspired by the success of previous work (Chattopadhyay et al. (2023)) that improved model generalizability by perturbing the high spatial frequency region in the amplitude spectrum of the training data, we adapted their approach to include low-frequency perturbations and uniform perturbations.

To perturb the clean images, we introduce $\epsilon_{uniform}$, $\epsilon_{high}[m,n]$ and $\epsilon_{low}[m,n]$. $\epsilon_{high}[m,n]$ and $\epsilon_{low}[m,n]$ are drawn from a Normal distribution, using the spatially dependent functions to control the perturbation extent. $\epsilon_{uniform}$ is spatially independent.

$$\epsilon_{high}[m,n] \sim \mathcal{N}\left(1, (2\alpha\sqrt{\frac{m^2+n^2}{H^2+W^2}}+\beta)^2\right), \tag{9}$$

$$\epsilon_{low}[m,n] \sim \mathcal{N}\left(1, (2\alpha\sqrt{\frac{(H-m)^2+(W-n)^2}{H^2+W^2}}+\beta)^2\right), \tag{10}$$

$$\epsilon_{uniform} = \alpha\beta, \tag{11}$$

where $\alpha \in \mathbb{R}, \alpha \in [3,5]$, $\beta = 0.3$, $m \in [-H/2, H/2]$, $n \in [-W/2, W/2]$. We then zero-centre the amplitude spectrum $\mathcal{A}(\boldsymbol{x})$ before applying the perturbation function $g(.)$

$$\begin{aligned}\hat{\mathcal{A}}(\boldsymbol{x}) &= g(\mathcal{A}(\boldsymbol{x}), \epsilon)[m,n]\\ &= \epsilon + \mathcal{A}(\boldsymbol{x}),\end{aligned} \tag{12}$$

where $\epsilon \in [\epsilon_{uniform}, \epsilon_{high}, \epsilon_{low}]$. Finally, we obtain the augmented image via 2D-FFT by recombining the perturbed amplitude spectrum with the original phase spectrum.

**FrequencyMix.**
While we have addressed the difficulty of modulating the data augmentation strength as a function of spatial frequency, we are still faced with the challenge of generating sufficiently diverse data for training the Y-FPN to condition the diffusion model.

The lack of relevant training examples under the test-time adaptation setting poses a considerable chal-

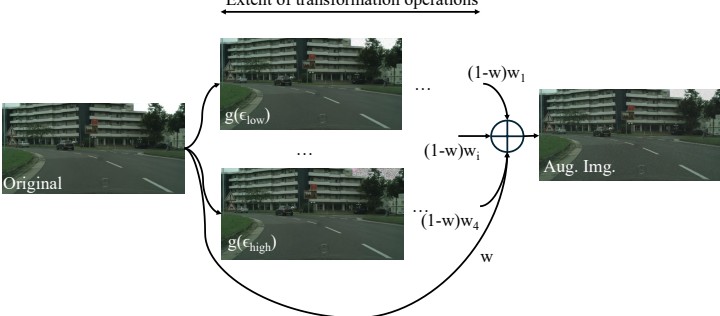

**Figure 2: Illustration of FrequencyMix**. We combine randomly selected augmentations and re-weight the augmented images to generate a composite augmented image. The weights $w_i$ are sampled from a Dirichlet distribution, and the blending weight $w$ is sampled from a Beta distribution.

lenge to training a robust model that is adapted to the target domain data. Previous work demonstrated the effectiveness of data augmentation by randomly sampling and combining data transformation operations for robust image classification (Hendrycks et al. (2020)) and image de-raining

(Guo et al. (2021)). However, we are faced with the daunting task of learning pixel-wise frequency filters that can extract important semantic information to condition the diffusion model while being simultaneously robust to noise from a broad range of frequencies.

Our design of FrequencyMix is motivated by an earlier observation from Yin et al. (2019) that improving model robustness often involves a *trade-off*. While data augmentation corresponding to a particular frequency range improves robustness to noise from that range, it also worsens robustness to noise from the excluded frequencies. This motivated our design and application of FrequencyMix to enhance the robustness of Y-FPN.

---

**Algorithm 1** FrequencyMix and Y-FPN training

---

**Input:** Clean Images $\mathcal{I}$, Y-like Frequency Prediction Network Y-FPN(.),
  Loss function $\mathcal{L}$., Augmentations $Aug = [\text{Uni. Noise}, g(\epsilon_{low}), g(\epsilon_{high}), g(\epsilon_{uniform})]$
**Output:** Trained Y-like Frequency Prediction Network Y-FPN(.)
  **Function** FrequencyMix(clean Image $\mathbf{I}$) :
 1: Sample mixing weights $(w_1, w_2, w_3, w_4) \sim Dirichlet$;
 2: Initialize an empty image $\mathbf{I}_{mix}$;
 3: **for** $i = 1, ..., 4$ **do**
 4:    Sample augmentations $(a_1, a_2, a_3) \sim Aug$;
 5:    Combine augmentations via $a_{12} = a_2 a_1$, $a_{123} = a_3 a_2 a_1$ and $a_{1234} = a_4 a_3 a_2 a_1$;
 6:    Sample operation $o \sim (a_1, a_{12}, a_{123}, a_{1234})$;
 7:    $\mathbf{I}_{mix} += w_i o(\mathbf{I})$;
 8: **end for**
 9: Sample a blending weight $w \sim Beta$;
10: **return** $\mathbf{I}_{noisy} = w\mathbf{I} + (1 - w)\mathbf{I}_{mix}$
11: **for** $i = 0, ..., iter_{num}$ **do**
12:    Sample a clean Image $\mathbf{I} \sim \mathcal{I}$;
13:    Generate noisy image with FrequencyMix $\mathbf{I}_{noisy} \leftarrow$ FrequencyMix($\mathbf{I}$);
14:    Generate estimate of clean image with Y-FPN $\hat{\mathbf{I}} \leftarrow$ Y-FPN($\mathbf{I}_{noisy}$);
15:    Calculate Loss $\mathcal{L}$ from eqn 8 and backpropagate ;
16:    Update Y-FPN(.) parameters.
17: **end for**

---

In this section, we further explore a novel solution, FrequencyMix, to address this challenge. We outline our approach in Algorithm 1. At each training iteration, we generate a noise image and add it to the sampled clean image from the training dataset. Our method carries out 4 separate operations comprising randomly sampled augmentations of varying length before recombining the transformed images to form a composite augmented image. We show a toy example in Fig. 2.

### 4.3 IMPLEMENTATION DETAILS

We train our Y-FPN (27.5M Parameters) on a NVIDIA RTX 3090 24GB for 50 epochs on the training data split across the datasets. Training time takes approximately 38 hours for Cityscapes. Similar to Hendrycks et al. (2020), we avoid data augmentations (*i.e.* color jitter, contrast and blur) that exist among the ImageNet-C corruptions. During training, we use Uniform Noise in addition to the augmentations introduced in Equation 12 instead of Gaussian Noise since Gaussian Noise is one of the test corruptions in the ImageNet-C. Note that Y-FPN does not require any guidance from the task model during training, making it applicable to task models with different architectures. Our inference time for FDD is 47 seconds/image while DDA is approximately 30-40 seconds.

## 5 EXPERIMENTAL DETAILS

In this section, we provide an overview of the datasets, task models and baselines. Lastly, we discuss the limitations of our approach.

**Datasets.** We use the semantic segmentation dataset Cityscapes (Cordts et al. (2016)) and ADE20k (Zhou et al. (2017; 2019)).

For depth estimation, we use the datasets NYU2K v2 (Silberman et al. (2012)) and Kitti (Geiger et al. (2013)). Finally, we also include the image classification dataset CIFAR-100 (Krizhevsky (2009)).

We benchmark our approach against distribution shifts by evaluating with corruptions (n=15) drawn from the categories (*i.e. Noise*, *Blur*, *Digital*, *Weather*) in ImageNet-C at severity level=5.

For evaluation metrics, we report the mean Intersection over Union (mIoU) for Cityscapes and mean Accuracy (mAcc) for ADE20k; Structure Similarity Index (SSIM) (Wang et al. (2004)) for depth estimation and top-1 accuracy for image classification.

**Baselines.** We compare our approach with following state of the art approaches:

- DDA (Gao et al. (2023)) utilizes a predefined image-level low-pass filter to condition a pre-trained ImageNet diffusion model for denoising of corrupted images. We use their official codebase for comparison.
- Diffusion TTA (Prabhudesai et al. (2023)) jointly fine-tunes a pretrained diffusion model and the task model for test time adaptation of corrupted images. We use the values reported in their paper for comparison.

**Task Models.** Our experiments are conducted with the following semantic segmentation models: DeepLabV3 (Chen et al. (2018)), which is a ResNet-based architecture (He et al. (2016)) and Seg-Former (Xie et al. (2021)). For depth estimation, we use MonoDepth2 (Godard et al. (2019)) and DenseDepth (Alhashim & Wonka (2018)). For image classification, we use the ResNet50 He et al. (2016) backbone and the ViT (Dosovitskiy et al. (2021)) backbone.

**Table 2:** Test Time Adaptation for depth estimation on NYU Depth v2 and semantic segmentation on ADE20k. "*" indicates reported values because of code unavailability. We use the metrics (Depth: SSIM, Segmentation: mAcc). We observe consistent improvement compared to DDA and competitive performance with Diffusion TTA for different image corruptions.

| | Gauss. | Fog | Frost | Snow | Contrast | Shot |
|---|---|---|---|---|---|---|
| Depth: DenseDepth | 79.10 | 72.70 | 81.60 | 81.30 | 77.40 | 72.20 |
| Diffusion TTA (Prabhudesai et al. (2023)) * | 82.10 (+3.00) | 74.10 (+1.40) | 84.40 (+2.80) | 82.10 (+0.80) | 77.40 (+0.00) | 73.00 (+0.80) |
| Depth: DenseDepth | 65.70 | 72.40 | 80.70 | 66.70 | 68.30 | 71.20 |
| DDA (Gao et al. (2023)) | 82.10 (+16.40) | 73.90 (+1.50) | 77.70 (-3.00) | 82.20 (+15.50) | 75.50 (+7.00) | 75.30 (+4.10) |
| Ours (FDD) | 82.20 (+16.50) | 76.00 (+3.60) | 80.80 (+0.10) | 82.60 (+15.90) | 76.60 (+8.30) | 76.00 (+4.80) |
| Segmentation: Segformer | 65.30 | 63.00 | 58.00 | 55.20 | 65.30 | 72.20 |
| Diffusion TTA (Prabhudesai et al. (2023)) * | 66.40 (+1.10) | 65.10 (+2.10) | 58.90 (+0.90) | 56.60 (+1.40) | 66.40 (+1.10) | 63.70 (+4.00) |
| Segmentation: Segformer | 19.42 | 51.79 | 28.40 | 30.65 | 40.28 | 20.40 |
| DDA (Gao et al. (2023)) | 41.14 (+21.72) | 30.39 (-21.40) | 23.45 (-4.95) | 21.27 (-9.38) | 29.54 (-10.74) | 41.21 (+20.81) |
| Ours (FDD) | 41.16 (+21.74) | 34.09 (-17.70) | 26.07 (-2.33) | 23.59 (-7.06) | 34.09 (-6.19) | 42.49 (+22.09) |

**Table 3:** Test Time Adaptation for depth estimation on Kitti and segmentation on Cityscapes. "N" refers to the number of reverse denoising steps during diffusion-driven denoising.We use the metrics (Depth: SSIM, Segmentation: mIoU).

| | Noise | | | Blur | | | | Weather | | | | Digital | | | | |
|---|---|---|---|---|---|---|---|---|---|---|---|---|---|---|---|---|
| | Gauss. | Impulse | Shot | Defocus | Glass | Motion | Zoom | Fog | Frost | Snow | Bright. | Contrast | Elastic | JPEG | Pixel. | Avg. |
| Depth: Monodepth | 43.13 | 42.69 | 45.28 | 49.69 | 41.60 | 60.31 | 80.76 | 68.12 | 74.45 | 84.91 | 87.26 | 56.54 | 87.31 | 79.42 | 82.03 | 65.57 |
| DDA (N=50) | 83.01 | 82.39 | 84.92 | 49.12 | 42.93 | 44.40 | 66.07 | 40.44 | 51.30 | 62.37 | 79.93 | 54.28 | 85.70 | 79.61 | 83.47 | 65.99 |
| Ours (N=50) | 80.54 | 79.87 | 83.14 | 49.77 | 43.57 | 59.32 | 78.90 | 67.56 | 72.82 | 81.54 | 86.51 | 42.07 | 86.93 | 80.15 | 80.83 | 71.57 |
| Ours (N=30) | 57.62 | 59.08 | 57.12 | 50.84 | 43.99 | 59.60 | 79.28 | 68.13 | 73.17 | 84.73 | 86.83 | 49.62 | 87.04 | 80.24 | 80.79 | 67.87 |
| Segmentation: DeeplabV3-R101 | 5.71 | 5.87 | 6.06 | 58.64 | 46.83 | 52.38 | 16.82 | 65.25 | 12.93 | 8.25 | 70.34 | 26.92 | 72.19 | 28.46 | 74.70 | 36.76 |
| DDA (N=50) | 45.05 | 45.14 | 48.55 | 49.39 | 52.79 | 41.22 | 7.03 | 18.48 | 3.48 | 13.87 | 38.73 | 1.92 | 54.39 | 39.36 | 56.67 | 34.40 |
| Ours (N=50) | 52.73 | 54.57 | 60.27 | 52.33 | 56.37 | 49.70 | 11.88 | 54.92 | 9.02 | 5.94 | 59.69 | 5.86 | 65.79 | 47.92 | 70.42 | 43.83 |
| Ours (N=30) | 51.33 | 50.11 | 58.39 | 56.84 | 49.08 | 51.95 | 11.37 | 55.89 | 8.15 | 4.15 | 62.14 | 4.30 | 68.21 | 46.28 | 71.33 | 43.30 |

## 5.1 COMPARISON WITH SOTAs

We present results on depth estimation (NYU Depth v2) and semantic segmentation (ADE20k) in Table 2. We observe that: ❶ our approach outperforms DDA for both tasks across the evaluated corruptions, demonstrating the advantages of the proposed approach in enhancing robustness to test-time corruptions. ❷ our approach is competitive with Diffusion-TTA, outperforming it on depth estimation tasks across several corruptions (*e.g.* Gaussian Noise, Fog, Snow and Contrast). This is significant because our approach has faster inference times and also requires less computational resources during training and inference compared to Diffusion-TTA. Our approach performs less well compared to Diffusion TTA on naturalistic corruptions. However, in all the cases, our method demonstrates improvements compared to DDA, illustrating that our approach mitigates some of the reduced performance from diffusion-driven denoising.

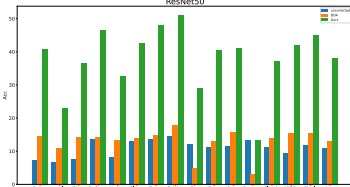 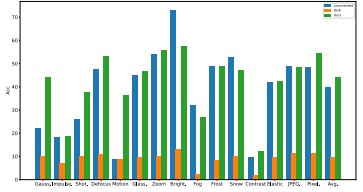

**Figure 3:** Image Classification (CIFAR-100c) for ResNet50 and ViT. For our approach, we condition the denoising process with the learned low-frequency filters.

To provide additional insights into the applicability of our method, we performed comparisons on additional datasets (Cityscapes, Kitti). From table 3), we observe that: ❶ our approach outperforms DDA on all the corruptions for both semantic segmentation and depth estimation except for "Noise", "Pixelate" on depth estimation and "Snow" for segmentation. For depth estimation, our approach improves performance with low-frequency corruptions compared to DDA (*e.g.* Motion Blur, Fog) while demonstrating competitive performance on high frequency noise (*e.g.* Gaussian Noise, Shot Noise) This further shows that our approach enhances the feasibility of diffusion-driven test time adaptation.

Since inference time is a critical consideration for test-time adaptation, we evaluate our approach at N=30 and N=50, where N is the number of reverse steps used during diffusion driven image denoising. Inference times for Cityscapes decreases from 47 seconds (N=50) to around 30 seconds per image (N=30), making test time adaptation more feasible. ❷ We observe that while performance may decrease as N decreases (*e.g.* "Noise"), our results remain competitive with DDA. Interestingly, decreasing N improved robustness to corruptions for both tasks (*e.g.* Fog, Motion Blur). Since "Noise" is characterized by high frequency noise, while "Fog" and "Motion Blur" are characterized by low-frequency noise, increasing N increases the the effectiveness of denoising for "Noise", while decreasing N reduces the smoothing effect and improves robustness to low-frequency noise.

To study the effect of spatially adaptive filters, we conducted a study on CIFAR-100c and only used the learned low frequency filters to condition the diffusion denoising process. Figure 3 shows that ❶ our approach outperforms DDA across all corruptions and model backbones, highlighting the effectiveness of spatially adaptive filtering, even for image-level classification tasks. ❷ We note that our approach improves performance relative to "Uncorrected" across model architectures with the exception of "Brightness" and "Snow" (ViT). However, in both cases, we observe that our approach improves performance compared to DDA.

Figure 4 provides some qualitative examples illustrating the performance of our approach on low-frequency ("Fog") and high-frequency ("Shot Noise").

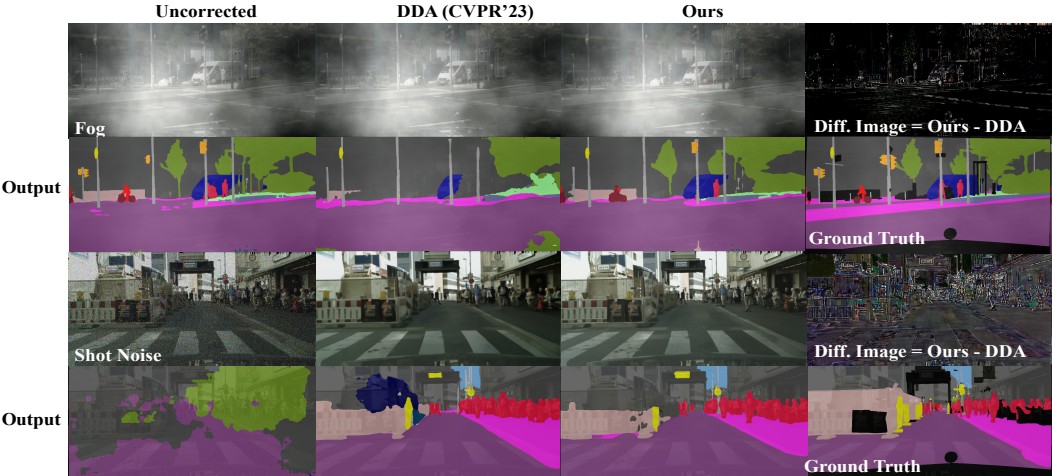

**Figure 4:** Our Frequency filter-Driven Diffusion approach jointly addresses the challenges posed during adaptation under low-frequency (*i.e.* fog) and high-frequency corruptions (*i.e.* shot-noise).

## 5.2 ABLATION STUDY

We present ablative analysis for the frequency filter based conditioning (Table 4). In Table 4, we observe that our learned low-pass filters (42.12) outperform the predefined image-level filters (41.40), further indicating the usefulness of spatially adaptive filters. Including the learned high-pass filters further improves the averaged performance (43.83), indicating the effectiveness of applying both high-pass and low-pass filters for dense classification. Additional results on ablation studies can be found in the Appendix.

## 5.3 LIMITATIONS

While our approach is competitive with state of the art diffusion-driven model- adaptation (Prabhudesai et al. (2023)) and input-adaptation methods (Gao et al. (2023)), we observe that our approach may sometimes perform less well compared to the uncorrected images.

**Table 4:** Ablation study (Cityscapes, DeeplabV3).

|  | mIoU |
| --- | --- |
| Baseline | 36.76 |
| + Predefined Image-Level filters | 41.40 |
| + Learned Low Pass filters | 42.12 |
| + Learned High Pass filters | 43.83 |

As noted by Gao et al. (2023), task performance on uncorrected images may be better than that of images with diffusion-driven denoising. They observed that diffusion models fail to restore images with naturalistic corruptions (*e.g.* "Fog", "Snow") because images with these corruptions occur in the ImageNet training data. Consequently, Gao et al. (2023) average the model prediction logits from the uncorrected image and the denoised images to mitigate the reduced performance. However, averaging logits is not possible for tasks such as depth estimation that do not involve logits. Furthermore, the effectiveness of self ensembling is dependent on the model's out of distribution performance. Notably, Gao et al. (2023) observed increasing performance gains on ImageNet-C with self ensembling as the task model increases in complexity (ResNet50:+1.3%, Swin-Tiny:+5.4%). Though strong out of distribution performance of the task model is essential for test time adaptation (Zhao et al. (2023)), we strived to develop a method that generalizes well to unseen data that reduces the need for a strong task model.

One potential solution is to introduce a noise-aware classifier (Luo et al. (2024)) that can characterize the noise, enabling the use of specialized denoising techniques that are tailored to the noise characteristics. However, such an approach would require extensive prior knowledge to train such a classifier. Recently, Oh et al. (2024) demonstrate the effectiveness of finetuning the diffusion model to improve robustness to image corruptions, which could also improve the performance of the denoising approach to naturalistic corruptions.

## 6 CONCLUSION

We show that our approach, Frequency-driven Diffusion (FDD) Adaptation effectively improves mean robustness over a broad range of noise corruption for input adaptation of dense classification tasks and different model architectures (CNN and transformer-based architectures). We show that our approach improves performance over previous diffusion-based input adaptation work, increasing its applicability beyond image-level classification tasks. Additionally, with the introduction of an easily trained module, our approach demonstrates improved robustness to diverse corruptions across different tasks, highlighting the effectiveness of our frequency-driven diffusion approach for input adaptation. Future work will also explore the possibility of extending our work towards purifying adversarially perturbed images.

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

# A  APPENDIX

In the appendix, we include further details on our work.

- A.1 More Visualization Results
- A.2 of denoising steps for FDD(Frequency Filter Driven Diffusion) for Test Time Adaptation
- A.3 Hyperparameter Study on Loss terms for Y-FPN
- A.4 Quantitative comparison of denoised image quality
- A.5 Comparison of FrequencyMix with baseline augmentation method

## A.1  MORE VISUALIZATION RESULTS

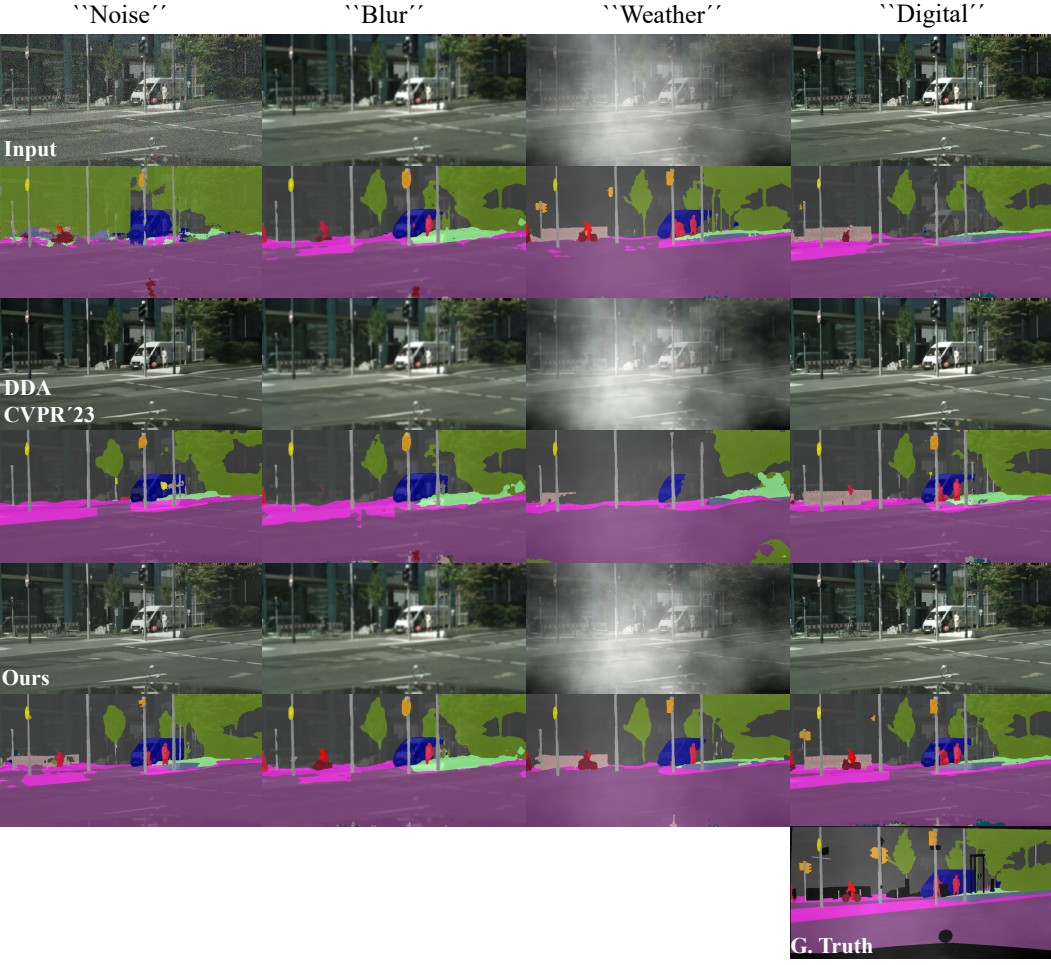

**Figure 5:** Qualitative comparison with state-of-the-art input- adaptation (*i.e.* DDA Gao et al. (2023)) and the original uncorrected input (*i.e.* Input) for the main corruption types-"Noise" (Impulse Noise), "Blur" (Defocus Blur), "Weather" (Fog) and "Digital" (Pixelate).

## A.2  NUMBER OF DENOISING STEPS FOR FDD(FREQUENCY FILTER DRIVEN DIFFUSION) FOR TEST TIME ADAPTATION

Since diffusion models have relatively long inference times, we were interested in determining whether conditioning the model with existing semantic information could reduce the minimum number of denoising steps. We found the performance decreases slightly when $N = 30$ and decreases markedly at $N = 10$ and $N = 60$.

**Table 5:** Number of denoising steps for the Cityscapes-C dataset. Adaptation performance peaks at $N = 50$.

| $N_{steps}$ | 10 | 30 | 50 | 60 |
|---|---|---|---|---|
| mIoU | 34.83 | 43.30 | 43.83 | 39.80 |

### A.3 HYPERPARAMETER STUDY ON LOSS TERMS FOR Y-FPN

We found that SSIM appears to have a stronger effect on performance compared to L1. Ultimately, we used the loss weights that provide robustness to both low-frequency and high-frequency corruptions.

**Table 6:** Hyperparameter study for loss terms L1, SSIM and $\mathcal{L}_{Freq}$ on Cityscapes-C (mIoU).

| L1 | SSIM | $\mathcal{L}_{Freq}$ | Gauss. | Motion. | Fog | Pixelate |
|---|---|---|---|---|---|---|
| 1.0 | 0.2 | 0 | 51.13 | **51.26** | 52.55 | 70.59 |
| 1.0 | 0.2 | 0.1 | 52.73 | 49.70 | 54.92 | 70.42 |
| 1.0 | 0 | 0 | 53.30 | 50.85 | **58.71** | **70.66** |
| 0 | 0.2 | 0 | 52.43 | 50.19 | 57.28 | 70.40 |
| 1.0 | 0.1 | 0 | **53.83** | 49.71 | 56.94 | 69.93 |

### A.4 QUANTITATIVE COMPARISON OF DENOISED IMAGE QUALITY

Our approach outperforms DDA across all categories except for "Weather" (PSNR).

**Table 7:** Quantitative comparison of denoised image quality relative to uncorrected images (Cityscapes-C).

| | Noise | | Blur | | Weather | | Digital | |
|---|---|---|---|---|---|---|---|---|
| | PSNR ↑ | SSIM↑ | PSNR ↑ | SSIM↑ | PSNR ↑ | SSIM↑ | PSNR ↑ | SSIM↑ |
| DDA Gao et al. (2023) | 13.011 | 0.732 | -1.658 | -0.048 | **1.213** | 0.002 | -3.257 | -0.060 |
| Ours (FDD) | **14.940** | **0.790** | **-1.257** | **-0.028** | 0.929 | **0.020** | **-0.805** | **0.009** |

### A.5 COMPARISON OF FREQUENCYMIX WITH BASELINE AUGMENTATION METHOD

**Table 8:** FrequencyMix improves performance (mIoU) relative to PASTA (Chattopadhyay et al. (2023)) on Cityscapes-C.

| | Gaussian Noise | Motion Blur | Fog | Pixelate |
|---|---|---|---|---|
| FrequencyMix | +33.9 | +39.4 | +50.8 | +45.4 |