# OpenReview forum: "Beautifying Diffusion Models: Learning Context-Aware Filters for Robust Dense Prediction on Test-Time Corrupted Images"
_ICLR.cc/2025/Conference — ICLR 2025 Conference Withdrawn Submission_

### Official Review · Reviewer_Fe8Y · 2024-10-28

**Soundness:** 2
**Presentation:** 3
**Contribution:** 2
**Rating:** 5
**Confidence:** 4

**Summary:**

This work introduces a diffusion-driven approach to Test time adaptation for dense prediction tasks. The proposed Frequency Driven Diffusion (FDD) harnesses both high and low-frequency information to preserve shape and color details during adaptation. The framework incorporates a Y-shaped frequency prediction network to apply context-sensitive low-pass and high-pass filters based on corrupted inputs. Moreover, the authors proposed FrequencyMix to create clean-corrupt image pairs to improve training robustness.

**Strengths:**

1. The proposed method is straightforward and easy to follow.

2. The motivation behind the Frequency Driven Diffusion (FDD) seems to make sense.

**Weaknesses:**

I have some questions about this paper that need further discussion. Please see them below.

If the authors can address my concerns, I am willing to raise my score.

**Questions:**

### Performance
 1. I noticed a significant performance drop for the proposed method in common real-world settings, such as fog and snow, with decreases of -17.70 and -7.06, respectively~(Table 2). Could the authors explain the reasons for this? Does this indicate a limitation in the model's generalization ability, suggesting it may only be effective in controlled scenarios like Contrast Shot?

### Baseline Selection
2. The proposed method seems like a "back-to-source" approach, aiming to denoise images for network training. However, methods such as TENT[1], CoTTA[2], and SVDP[3] focus on adapting the model to the target domain. I hope the authors can compare with these methods or at least discuss with this category of techniques in related work.

### Benchmark
3. Can the proposed method generalize to the widely used TTA dense prediction benchmark like ACDC?

[1] Tent: Fully Test-time Adaptation by Entropy Minimization
[2] Continual Test-Time Domain Adaptation
[3] Exploring Sparse Visual Prompt for Domain Adaptive Dense Prediction

---

### Official Review · Reviewer_e74m · 2024-11-02

**Soundness:** 3
**Presentation:** 2
**Contribution:** 3
**Rating:** 5
**Confidence:** 3

**Summary:**

This paper proposes a frequency-driven diffusion model （Frequency-Driven Diffusion, FDD） to improve the robustness of dense prediction tasks (such as semantic segmentation and depth estimation). In particular, the method proposed in this paper performs well on image corrosion when processing tests. FDD predicts pixel-level high-frequency and low-frequency filters through a Y-shaped frequency prediction network (Y-FPN) to better preserve image details. On this basis, the authors designed a FrequencyMix data augmentation method to generate training data adapted to noises of different frequencies, thereby improving the adaptability of the model under various corrosion types. Experimental results show that FDD outperforms existing methods on multiple datasets and maintains high performance under most corrosion conditions, although there is still room for improvement in the handling of natural corrosion effects.

**Strengths:**

1.This paper proposes a new frequency-driven diffusion model FDD, which predicts upper and lower frequency filters through the Y-shaped frequency prediction network (Y-FPN) to denoise noise of different frequencies, thereby enhancing the robustness of dense prediction tasks.

2.The paper structure is clear and conforms to the standards of academic papers. From the abstract, introduction, related work, methods, experimental details to the conclusion and references, each section is complete and logically coherent.

**Weaknesses:**

1.The inference phase takes a long time, especially in complex scenes (such as Cityscapes), where the inference time is 47 seconds per image. Although reducing the number of steps can shorten the time, it will affect the performance of some scenes, which may not be ideal in practical applications.

2.Although this paper performs well in most noise types, it has poor effects on natural corrosion effects (such as fog, snow, etc.).

3.Although Y-FPN can learn upper and lower frequency filters, this pixel-based filter learning increases the complexity of the network structure and places higher requirements on computing resources.

4.Although the overall logic is clear in the explanation of some technical details, the explanation of certain key concepts and operations can be more in-depth and detailed. For example, in the Frequency Mix training method, the specific effects and interrelationships of different frequency disturbances can be further explained to help readers better understand their principles and their impact on model training.

**Questions:**

1.Explanations of certain key concepts and operations can be more in-depth and detailed. For example, in the Frequency Mix training method, the specific effects and interrelationships of different frequency disturbances can be further explained to help readers better understand their principles and their impact on model training.

2.It should be explained the inference time and time complexity in practical applications.

3.The reason for the poor treatment effect on natural corrosion has not been explained.

---

### Official Review · Reviewer_KmLW · 2024-11-04

**Soundness:** 2
**Presentation:** 3
**Contribution:** 2
**Rating:** 5
**Confidence:** 4

**Summary:**

This paper proposes an input adaptation method FDD for dense prediction tasks by guiding a unconditional diffusion model to map test-time, potentially corrupted images back to the clean image domain. To perform the guidance, the paper observes that preserving a combination of low frequency and high frequency information is in general helpful, and the optimal combination can differ for different spatial locations. The paper thus learns a kernel prediction network that predicts a low pass filter and a high pass filter for each pixel location, and it proposes a FrequencyMix technique to train this network. Evaluation is performed on depth estimation, segmentation and classification tasks with a variety of task models, and with comparison with various existing methods.

**Strengths:**

* The presentation of the paper is clear and easy to follow.
* The evaluation is comprehensive, encompassing a variety of tasks and architectures. The ablation studies show the effect of each component of the proposed method.
* The proposed approach, which does not require target domain data or finetuning the diffusion or task models, is interesting and elegant.

**Weaknesses:**

* The main limitation is that based on Table 2 and 3, although FDD is very effective for noise corruptions, it does not consistently improve upon the uncorrected inputs for other types of corruptions (blur/weather/digital): In Table 2 Segformer on ADE20k, FDD's performance on the weather and contrast distortions is below that of no input adaptation. In Table 3, its average performance within the blur, weather, and digital corruption categories generally seems to be lower than that of no input adaptation. In various cases, the performance decrease is quite significant. This limits the overall usefulness and reliability of FDD, as noise corruption only represents a limited range of the corruptions that can be encountered in real-world scenarios. Also, given that applying FDD on certain types of corruption may reduce performance, there could be additional complexity at test time for needing to identify what kind of corruption an input may have and decide whether to apply FDD to it or not.
* Additionally, FDD takes 30-47 secs per image for inference, which is relatively slow. This makes it hard to apply FDD on many application tasks where real-time performance is important.

**Questions:**

* Please see weaknesses above.
* Also, a potential missing ablation/analysis is performance vs. different corruption levels. When a test-time input is already close to the clean, training distribution, would FDD be able to correctly handle it and make minimal change to it?

---

### Note · Authors · 2024-11-13

**Comment:**

Dear Reviewers KmLW, e74m and Fe8Y,

thank you all for your helpful comments on the above manuscript and we will work towards making it better.
While my coauthors and I will be withdrawing this submission, we hope to submit to ICLR again in the near future.

Thank you all for your kind attention.

**Withdrawal Confirmation:**

I have read and agree with the venue's withdrawal policy on behalf of myself and my co-authors.